# Square Root Unscented Kalman Filter-Based Multiple-Model Fault Diagnosis of PEM Fuel Cells

**DOI:** 10.3390/s25010029

**Published:** 2024-12-24

**Authors:** Abdulrahman Allam, Michael Mangold, Ping Zhang

**Affiliations:** 1Institute of Applied Mathematics, Bingen University of Applied Sciences, 55411 Bingen am Rhein, Germany; m.mangold@th-bingen.de; 2Institute of Autmatic Control, University of Kaiserslautern-Landau, 67653 Kaiserslautern, Germany; pzhang@eit.uni-kl.de

**Keywords:** proton exchange membrane fuel cells, flooding, catalytic degradation, multiple-model fault diagnosis, square root unscented kalman filter

## Abstract

Harsh operating conditions imposed by vehicular applications significantly limit the utilization of proton exchange membrane fuel cells (PEMFCs) in electric propulsion systems. Improper/poor management and supervision of rapidly varying current demands can lead to undesired electrochemical reactions and critical cell failures. Among other failures, flooding and catalytic degradation are failure mechanisms that directly impact the composition of the membrane electrode assembly and can cause irreversible cell performance deterioration. Due to the functional significance and high manufacturing costs of the catalyst layer, monitoring internal fuel cell states is crucial. For this purpose, a diagnostic-oriented multi-scale PEMFC catalytic degradation model is developed which incorporates the failure effects of catalytic degradation on cell dynamics and global stack performance. Embedded to the multi-scale model is a square root unscented Kalman filter (SRUKF)-based multiple-model fault diagnosis scheme. In this approach, multiple models are used to estimate specific internal PEMFC system parameters, such as the mass transfer coefficient of the gas diffusion layer or the exchange current density, which are treated as additional system states. Online state estimates are provided by the SRUKF, which additionally propagates model-conditioned statistical information to update a Bayesian framework for model selection. The Bayesian model selection method carries fault indication signals that are interpreted by a derived decision logic to obtain reliable information on the current-operating system regime. The proposed diagnosis scheme is evaluated through simulations using the LA 92 and NEDC driving cycles.

## 1. Introduction

Reducing greenhouse gas emissions through the employment of alternative energy solutions is a worldwide challenge. In 2020, Germany released the first issue of the National Hydrogen Strategy defining a roadmap to achieve the national set objective of carbon neutrality in the year 2045 [1]. The document recognizes green hydrogen as a leading energy solution in the decarbonization of the transportation sector. Specifically, it is suggested to employ hydrogen as a main fuel for the road freight industry and to promote the deployment of heavy-duty hydrogen fuel cell electric vehicles (HFCEVs). Due to favorable characteristics such as quick start-ups, high energy density, and moderate operating temperatures, PEMFCs are highly applicable to electric propulsion systems. However, cell durability poses a technological limitation when competing with conventional powertrains such as internal combustion engines. Currently, stack durability in heavy-duty vehicles ranges around 20,000 h with a set target to reach 30,000 h by 2030 [2].

Harsh operational conditions imposed by vehicular applications, such as cold start-ups, load cycling, and external environmental conditions, significantly contribute to the limited durability of PEMFCs. In comparison to stationary applications, experimental results demonstrate that stack voltage decay is significantly higher in applications with transient operating conditions [3]. Among other factors, suboptimal conditions and uncontrolled hazardous electrochemical reactions in the membrane electrode assembly (MEA) are a major contributor to this performance decay. However, obtaining in situ measurements from the MEA is infeasible due to the compactness and sensitivity of the assembly. Therefore, integrating advanced diagnosis strategies to monitor critical system parameters for the early detection of PEMFC failures is imperative to enhance cell durability.

Flooding and catalytic degradation are cell failures that directly impact the functionality and integrity of the MEA. Flooding is a consequence of improper water management where excess liquid water accumulates in the gas channels, the gas diffusion layer, and in the porous structure of the electrode. Different degrees of flooding increase mass transportation resistances and dilute reactant concentrations promoting sub-stoichiometric electrochemical reactions. Additionally, the presence of excess water at high potentials induces catalytic degradation through the mechanism of carbon corrosion which has negative long-term effects on cell performance [4].

Catalytic degradation is an irreversible mechanism that directly impacts the chemical composition of the electrode and is mainly attributed to the loss of electrochemically active surface area (ECSA). Under suboptimal operating conditions, the carbon-supported platinum particles dissolute and migrate to neighbouring platinum particles or, in the presence of water, oxidize to form a platinum oxide layer [5,6]. Overall, morphological changes in the catalyst layer disrupt stack performance and can lead to premature stack failure. Due to the high costs of platinum and the significance of the catalyst layer in cell operations, a multi-scale PEMFC catalytic degradation model is developed to obtain a qualitative yet accurate representation of the molecular effects of catalytic degradation on overall cell performance.

Additionally, to test the fault diagnosis system, reference data are required. Experimental data of degrading fuel cells in vehicles are hard to obtain. Therefore, simulated data from a reference model are used in this work. The reference model is a multi-scale PEMFC degradation model which has not been published before and is suggested as a testing environment for diagnosis systems. The model incorporates the mechanisms of platinum dissolution and platinum oxide formation.

Due to the complexity and variety of degradation mechanisms in PEMFCs, it is inevitable that the reference model may deviate from the true degradation in PEMFCS. It should be considered as one possible degradation scenario among others. Consequently, the developed fault diagnosis scheme should not be focused on a specific reference model but kept more general in order to detect degradation mechanisms that are not included in the reference model. To achieve this objective, the PEMFC models in the diagnosis system are designed to be more general than the reference model. A plant–model mismatch is intentionally included in the development of the diagnosis scheme.

In the literature, model-based and non-model-based approaches have been adopted to diagnose flooding and catalytic degradation. Signal processing is a widely established non-model-based fault diagnosis strategy where mathematical transformations of system measurements are utilized for signal feature extractions and fault pattern recognition. In [7,8], the discreet wavelet transformation is implemented to recognize characteristic patterns of a voltage signal under the failure effects of flooding. Similarly, reference [9] experimentally induces flooding by manipulating stack relative humidity and detects flooding by extracting features from a voltage signal using empirical mode decomposition. The work proposed in [10] extracts information from a pressure drop signal in the frequency domain which is correlated to cell voltage measurements to diagnose water levels in the stack. The authors introduce sudden step-up input currents to observe the temporal dynamics of excessively hydrated PEMFC systems in response to rapid load changes.

Machine learning approaches have been also investigated in the fault diagnosis of PEMFCs. The work in [11] processes datasets of voltage measurements using the Fisher discriminatory analysis and a support vector machine (SVM) classifier to diagnose the state of health of a PEMFC stack. The diagnosis of water management failures using principal component analysis and SVM classifiers is investigated in [12]. In [10], the authors present a comprehensive overview on criteria for the evaluation of data-driven approaches in PEMFCs. Additionally, the authors investigate different classification algorithms trained from data generated by an electrochemical impedance spectroscopy (EIS) to diagnose different degrees of cell flooding. Data-driven approaches haven been applied successfully. Compared to approaches based on physical models, data-driven models offer the advantage of avoiding the expensive step for model development. On the other hand, a vast physical and electrochemical knowledge on the processes in fuel cells has been accumulated over the years. Ignoring this knowledge may lead to diagnosis systems with minor informative values and limited extrapolation capabilities. The challenge is to balance between the physical knowledge included into a diagnosis system on the one hand and the costs for model development on the other hand.

A rather simple way to model electrochemical systems are linear equivalent circuit models. Experimental approaches such the EIS are commonly proposed in the literature as a tool for the parameter identification of equivalent electrical circuit models [13]. Deviations of circuit parameters from predefined nominal values indicates faulty operating modes and anomalies in the PEMFC system [14,15]. The authors in [16] utilize data generated from an EIS to measure impedances of an equivalent circuit model during an accelerated stress test on a cathodic catalyst layer. However, the diagnosis performance of EIS in real-time applications, typical for vehicular applications, is limited due to the prerequisite of steady-state operating conditions during the sweep [17]. Additionally, equivalent circuits fail to accurately represent the temporal dynamics of a PEMFC system leading to loss of information and model uncertainties [18].

A more detailed model-based approach is presented in [19] to detect flooding by developing a first principles fuel cell model. The authors incorporate the effects of flooding as a parametric change impacting the air flux in the gas flow channels. In [20], a parameter estimation approach is adopted to detect different degrees of flooding. The authors model the volume of liquid water in the flow channels and the GDL as system parameters that are jointly estimated using the unscented Kalman filter (UKF) and are implemented as indicators of water levels in the MEA. Catalytic degradation is modelled in [21,22] as an abrupt change in the ESCA. The diagnosis system is based on analytical redundancy and the processing of residual signals. In a previous publication, the performance of the SRUKF and the particle filter with a sample size of 200 particles were compared within a multiple-model scheme. Conclusively, the SRUKF exhibited an overall better performance with a much lower computational overhead [23]. Recent publications on fault diagnosis on PEMFC are focused on data-driven approaches.

Based on the literature review analysis and motivated by recent developments in fault diagnosis [24], the main contributions are offered in this paper:To develop a diagnostic oriented multi-scale PEMFC catalytic degradation model that investigates the failure mechanisms of catalytic degradation through platinum dissolution and platinum oxide formation [25] on overall stack performance. The model can be used to generate reference data to test the proposed fault diagnosis system in effectively detecting catalytic degradation in PEM fuel cells.To implement the principles of multiple-model fault diagnosis, first introduced in [26], combined with the SRUKF as a Bayes’ filter for the diagnosis of fuel cell systems in dynamic operations [27]. Fundamentally, the main task of multiple-model fault diagnosis schemes is to select the underlying model which most likely represents the current system state.

In a previous publication, the performance of the SRUKF and the particle filter with a sample size of 200 particles was compared within a multiple-model scheme. Conclusively, the SRUKF exhibited an overall better performance with lower computational overhead [23]. Recent publications focus on data-driven approaches for the fault diagnosis of PEM fuel cells. While data-driven approaches can be effective in diagnosing flooding and catalytic degradation, little knowledge can be obtained from the internal states of the stack which are important to plan and implement preventive and mitigation actions. Furthermore, the real-time implementation of data-driven approaches on complex applications requires additional online learning strategies, which increases the complexity of application and computational costs [28].

The paper is organized as follows. A multi-scale PEMFC catalytic degradation model driven by a simplified vehicle model is presented in Section 2.1. In Section 2.3, an analysis is conducted on the fault modelling process of flooding and catalytic degradation. The developed multiple-model fault diagnosis scheme is presented in Section 2.4. Simulation results are presented in Section 3 to validate the proposed scheme using the LA92 and the NEDC driving cycle.

## 2. Materials and Methods

### 2.1. Multi-Scale PEMFC Catalytic Degradation Model

The Membrane Electrode assembly (MEA) is a crucial part of PEMFCs consisting of a polymer membrane sandwiched between an anode and a cathode. Both electrodes are symmetrical and are composed of two layers, namely the gas diffusion layer and the catalyst layer. The gas diffusion layer acts as a transport medium to homogeneously distribute the reactants along the surface area of the catalyst layer. The catalyst layer is designed to provide optimal conditions for low-temperature redox reactions using finely distributed carbon supported platinum particles. Among other factors, cell performance deterioration and stack durability are strongly attributed to the state of health of the catalyst layer and are impacted by suboptimal operating conditions in the MEA.

Specifically, the catalyst layer is prone to chemical degradation where the composition of the carbon supported platinum particle is disrupted. In the literature, comprehensive physical models of the mechanisms of catalytic degradation such as platinum dissolution, platinum oxide formation and reduction, platinum disposition, and carbon corrosion are developed in [6,29,30]. Further works presented in [31,32,33] expand on the degradation models by investigating the effects of operating conditions that are relevant to automotive applications on individual degradation mechanisms and degradation rates. Additionally, comprehensive multi-scale models that couple the local nanoscale effects of catalytic degradation on global overall stack performance are developed in [34,35,36]. However, the presented models are highly detailed and are less suitable for the development of a diagnosis tool. Therefore, a refined diagnostic oriented model is developed in this paper based on [25] to incorporate the effects of catalytic degradation. The core of the multi-scale model is proposed in a previous publication [23]. It is presented here in more detail. Further, the model is embedded in a simple vehicle model in order to simulate the fuel cell system under vehicular operating conditions.

The multi-scale PEMFC degradation model is developed with two main objectives. Firstly, the model provides a qualitative understanding of the molecular effects of catalytic degradation on the overall stack performance. In addition, the multi-scale model generates reference measurement data to evaluate the performance of the proposed diagnosis system. The following is assumed to develop the multi-scale model:Gases behave like ideal gases.PEMFC is an isothermal system.The anodic overpotential is neglected.All cells in the stack are homogeneous.The catalyst layer is composed of spherically shaped carbon supported platinum particles.Platinum dissolution and platinum oxide formation and reduction are the only mechanisms responsible for catalytic degradation.Values of the system parameters are listed in Appendix A.

#### 2.1.1. Macroscopic Model

The macroscopic part of the PEMFC system is described by a three-state half-cell semi-empirical model capturing the cathodic mass and charge balances. It has been previously validated in [37]. The nonlinear model is a multiple-input–multiple-output system with three states identified as the molar concentration of oxygen in the gas flow channels $x_{1}$, the molar concentration of oxygen in the catalyst layer $x_{2}$, and the cathodic overpotential $x_{3}$. The model inputs are the electric current $u_{1}$, the inlet air flow $u_{2}$, and the partial molar concentration of oxygen in the supplied air $u_{3}$. The two measurements are the molar concentration of oxygen in the catalyst layer $y_{1}$ and the cell voltage output $y_{2}$. The inlet air flow is fed through the channels grooved in the bipolar plates and is uniformly distributed through the GDL onto the catalyst layer. Assuming the GDL to behave as an ideal microporous diffusion media, the mass flux between both the flow channels and the catalyst layer is described by
(1)$$J_{O_{2}}=A_{E}\beta \left(x_{1}-x_{2}\right)$$
where β is the mass transfer coefficient of the GDL defining the permeability of the layer and $A_{E}$ is the mass exchange surface area. In the catalyst layer, oxygen encounters a catalytic environment and is reduced in an electrochemical reaction with a rate that is defined by the Butler–Volmer kinetics equation,
(2)$$i_{c}=k_{cd}i_{c,0}\left(\exp\left(\frac{\alpha_{o}Fx_{3}}{R_{gas}T}\right)-\exp\left(\frac{\left(1-\alpha_{o}\right)Fx_{3}}{R_{gas}T}\right)\frac{x_{2}}{x_{2,0}}\right)$$
where *F* is the Faraday constant, $R_{gas}$ is the universal gas constant, *T* is the cell temperature, $i_{c,0}$ is the exchange current density characterizing the activeness of the catalyst layer, and $x_{2,0}$ is a reference molar concentration. The parameter $k_{cd}$ defines the rate of catalytic degradation, which is comprehensively covered at the end of this section. Implementing mass conservation principles, the molar concentration of oxygen in the gas flow channel is defined as
(3)$${\dot{x}}_{1}=\frac{1}{V_{C}}\left(u_{2}\left(u_{3}-x_{2}\right)-J_{O_{2}}\right),$$
with $V_{C}$ as the gas bulk volume. The molar concentration of oxygen in the catalyst layer is a function of the mass flux and the reaction rate of oxygen,
(4)$${\dot{x}}_{2}=\frac{1}{V_{CL}}\left(J_{O_{2}}+\frac{A_{E}}{4F}i_{c}\right)$$
where $V_{CL}$ is the volume of the gas pores in the catalyst layer. The third state describes the activation losses which quantifies the energy dissipated to break the ionic bonds to initiate electrochemical reactions. From a charge balance, the dynamics of the system state is captured by
(5)$${\dot{x}}_{3}=\frac{1}{\phi_{C}}\left(-u_{1}-\frac{A_{E}}{F}i_{c}\right).$$

The activation losses are a function of the charge double layer $\phi_{C}$, identified as a capacitance effect generated by an inter facial ionic accumulation between the polymer membrane and the catalyst layer.

#### 2.1.2. Microscopic Model

In this section, the microscopic model is developed to incorporate the failure effects of catalytic degradation on overall cell dynamics and global stack performance. The loss of ECSA is considered as the main cause for the degradation. On a molecular level, the mechanisms of catalytic degradation can be physically represented by the interfacial electrochemical reactions of platinum dissolution (Equation 6), platinum oxide formation and reduction (Equation 7), and the chemical dissolution of platinum oxide (Equation 8),
(6)$$\mathrm{Pt}={\mathrm{Pt}}^{2+}+2e^{-},$$


(7)
$$\mathrm{Pt}+H_{2}O=\mathrm{PtO}+2H^{+}+2e^{-},$$



(8)
$$\mathrm{PtO}+2H^{+}={\mathrm{Pt}}^{2+}+H_{2}O.$$


The microscopic model is developed based on Reactions (Equation 6)–(Equation 8) and is based on the work proposed in [25]. In [25], the authors develop a 3-state spatially lumped platinum dissolution model assuming uniformly distributed carbon-supported spherically shaped platinum particles. The proposed model is modified with two simplifications. Firstly, the model is reduced to two states identified as the platinum oxide coverage of particle $x_{4}$, and the platinum particle radius $x_{5}$. The molar concentration of platinum ions in the solution is neglected due to its irrelevance in computing the coupling term $k_{CD}$ defined in (Equation 17). The second simplification is imposed as a worst-case degradation scenario characterized by a unidirectional forward reaction of platinum dissolution described in (Equation 6) while neglecting the reverse reaction of platinum plating. As a result, the Butler–Volmer kinetic equation of Reaction (Equation 6) is formulated as
(9)$$r_{1}=k_{1}\left(1-x_{4}\right)\left[\exp\left(\frac{\alpha_{1}n_{1}F}{RT}\left(U_{CM}-U_{Pt}\right)\right)\right]$$
with $\left(1-x_{4}\right)$ as the unoxidized surface of the platinum particles available to engage in electrochemical reactions, $k_{1}$ is a reaction constant dictating the forward reaction rate of platinum dissolution, and $\alpha_{a,1}$ is the anodic mass transfer coefficient. The activation potential of Reactions (Equation 6) and (Equation 7) are influenced by the local potential difference across the cathode-membrane interface $U_{CM}$ which is equated as
(10)$$U_{CM}=U_{0}-x_{3},$$
and the standard equilibrium potential of platinum dissolution $U_{Pt}$. To include the effects of particle size, $U_{Pt}$ is shifted by the Kelvin term
(11)$$U_{Pt}=U_{1}^{\theta }-\frac{1}{2F}\left(\frac{\sigma_{Pt}M_{Pt}}{x_{5}\rho_{Pt}}\right)$$
where $U_{1}^{\theta }$ is the thermodynamically reversible of platinum dissolution. The magnitude of the Kelvin term depends on the surface tension of platinum particles $\sigma_{Pt}$, the molecular weight of platinum $M_{Pt}$, the density of platinum $\rho_{Pt}$ and the particle radius $x_{5}$. In the presence of water, the electrochemical reaction of platinum oxide (Equation 7) is initiated with a rate defined by
(12)$$\begin{array}{cc} r_{2}= & k_{2}\left[\exp\left(-\frac{\omega x_{4}}{RT}\right)\exp\left(\frac{\alpha_{2}n_{2}F}{RT}\left(U_{CM}-U_{PtO}\right)\right)\right.\\ \; & \left.-x_{4}\left(\frac{c_{H^{+}}^{2}}{c_{H^{+},ref}^{2}}\right)\exp\left(-\frac{\alpha_{3}n_{2}F}{RT}\left(U_{CM}-U_{PtO}\right)\right)\right] \end{array}$$
where $k_{2}$ is a reaction rate constant, ω is the platinum oxide interaction coefficient, $\alpha_{a,2}$ and $\alpha_{c,2}$ are, respectively, the anodic and cathodic mass transfer coefficients. With the inclusion of the Kelvin term for platinum oxide particles, the standard equilibrium potential for platinum oxide formation is equated as
(13)$$U_{PtO}=U_{2}^{\theta }+\frac{1}{2F}\left(\Delta \mu_{PtO}^{0}+\frac{\sigma_{PtO}M_{PtO}}{x_{5}\rho_{PtO}}-\frac{\sigma_{Pt}M_{Pt}}{x_{5}\rho_{Pt}}\right),$$
where $U_{2}^{\theta }$ is the thermodynamically reversible potential of Reaction (7) and $\Delta \mu_{PtO}^{0}$ is a fitted electrochemical potential of platinum oxide formation. $\sigma_{PtO}$, $M_{PtO}$ and $\rho_{PtO}$ are, respectively, the surface tension, the molecular weight, and the density of platinum oxide. The parameter $c_{H^{+}}$ defines the proton concentration as a function of membrane water content and in relation to a reference concentration value $c_{H^{+},ref}$. Although Reaction (Equation 8) can be obtained by balancing Reactions (Equation 6) and (Equation 7), a similar assumption to [25] is adopted where the chemical pathway of platinum oxide dissolution is thermodynamically independent. Subsequently, the reaction rate of (Equation 8) is calculated as
(14)$$r_{3}=k_{3}x_{4}c_{H^{+}}^{2}$$
where $k_{3}$ is a reaction rate constant.

Based on reaction rates $r_{1}$, $r_{2}$ and $r_{3}$, the temporal evolution of catalytic degradation is characterized by two states, the platinum oxide coverage $x_{4}$ and the platinum particle radius $x_{5}$ whose balance equations are expressed as
(15)$${\dot{x}}_{4}=\left(\frac{r_{2}-r_{3}}{\Gamma_{max}}\right)-\left(\frac{2x_{4}}{x_{5}}\right){\dot{x}}_{5}$$
(16)$${\dot{x}}_{5}=-\frac{M}{\rho }\left(r_{1}-r_{2}\right),$$
where $\Gamma_{max}$ is number of moles of active sites per platinum area. The multi-scale model is directly coupled using $k_{cd}$ which characterizes the degradation rate based on the ratio between the actual ECSA and the initial electrode ECSA of a newly manufactured cell,
(17)$$k_{CD}=\frac{x_{5}^{2}\left(1-x_{4}\right)}{r_{o}^{2}}$$

While there are other degradation mechanisms that impact the catalyst layer such as carbon corrosion and Ostwald ripening, including these mechanisms results in a highly complex microscopic model. Given that the proposed model is diagnostic-oriented, the main objective is to develop a model which can qualitatively describe the molecular effects of catalytic degradation with low computational time and serve as a reference data generator for the failure mechanism. Additionally, imposing a worst-case degradation mechanism is a feasible assumption as it simulates the effects of baseline degradation which occurs inevitably in PEM fuel cells over the long term [38].

The two observed measurements of the multi-scale model are the molar concentration of oxygen in the catalyst layer $y_{1}=x_{2}$ and the cell voltage output $y_{2}$. The cell voltage is computed by deducting activation and ohmic losses $R_{ohm}$ from the reversible open-circuit voltage $U_{0}$,
(18)$$y_{2}=U_{0}+x_{3}-R_{ohm}u_{1}.$$

### 2.2. Vehicle Model

In order to test the multi-scale model and to provide realistic load scenarios for the PEMFC system, the model is embedded to a very simple vehicle model. The purpose of the vehicle model is to generate electric current demands for the fuel cell. As this work is interested in degradation, a worst case scenario of catalytic degradation is assumed. The PEMFC is exposed to unfiltered load changes due to standard driving cycles. In reality, auxiliary systems such as tracking batteries or load balancing schemes would be integrated to at least partially protect the fuel cell. But again, the focus of this paper is on degrading PEM fuel cells and not on stack protection. A simple vehicle model is developed in this subsection to convert the velocity profile in of driving schedules to an input electric current for the multi-scale PEMFC degradation model. Power consumption due to road angles and power losses due to stack efficiency can be included in future work. For simplicity, the longitudinal vehicular forces can be computed by Newtons second law as
(19)$$m_{v}a=F_{t}\left(t\right)-F_{\mathrm{drag}}\left(t\right)-F_{\mathrm{roll}}\left(t\right)$$
where $F_{t}$ is the propulsion force required from the stack, $F_{\mathrm{drag}}$ are the aerodynamic resistances, and $F_{\mathrm{roll}}$ are the rolling resistances in the tires. At first, the given velocity profile of the driving cycle is converted to acceleration
(20)$$\vec{a}=\frac{d\vec{\nu }}{dt}.$$

In the next step, forces due to aerodynamic and rolling resistances are calculated as
(21)$$F_{\mathrm{drag}}=\frac{1}{2}\rho_{\mathrm{air}\;}A_{V}C_{D}v^{2}$$
(22)$$F_{\mathrm{roll}}=m_{v}g\mu_{\mathrm{Roll}\;}v$$
where $\rho_{air}$ is the density of air, $A_{V}$ is frontal area of the vehicle, and $C_{D}$ is the drag coefficient. In (22), *g* is gravitational constant and $\mu_{Roll}$ is a rolling resistance coefficient. Rearranging Equation (Equation 19), the total power demanded from the stack is computed as
(23)$$P_{\mathrm{stack}\;}=\left(m_{v}a+F_{\mathrm{drag}}\left(t\right)+F_{\mathrm{roll}}\left(t\right)\right)v$$

After computing the power demand from the velocity profile of the driving cycle, the electric current $u_{1}$ for a single cell is computed as an input for the developed PEMFC degradation model as
(24)$$u_{1}=\frac{P_{\mathrm{stack}\;}}{U_{0}n_{\mathrm{cells}\;}}$$
where $U_{0}$ is the open-circuit voltage and $n_{cells}$ is the number of cells in the stack.

### 2.3. Design of Fault Diagnosis Models

Developing a reliable model for a fault diagnosis scheme is not a trivial task. On the one hand, the model should be detailed enough to indicate physical causes for a failure. On the other hand, the fault diagnosis model should be sufficiently general to additionally capture unknown types of faults. For example, the multi-scale model of Section 2.1 describes catalytic degradation due to the mechanisms of platinum dissolution and platinum oxide formation quite accurately and hence may provide a test case for a diagnosis system. However, the multi-scale model does not contain other degradation mechanisms such as carbon corrosion. Including all possible physio-chemical mechanisms responsible for catalytic degradation in a fault diagnosis model is a formidable task that results in a model equation system too complex for online monitoring purposes. Therefore, another approach is adopted in this work. The division of the multi-scale model with clear interfaces between the macroscopic and microscopic sub-models shows a way towards a suitable diagnosis model. The idea is to only utilize the macroscopic model in the diagnosis system. Additionally, the fault mechanisms are no longer reflected in detail in the multiple models of the fault diagnosis scheme. Instead, the faults are captured by introducing a time-dependent system parameter with a clear physical interpretation. This is outlined in the following for the fault states of flooding and catalytic degradation.

Flooding and catalytic degradation are classified as multiplicative faults that directly disrupt system stability and are accurately modelled by changes in system parameters. Therefore, the failure analysis conducted in this section assists in determining relevant system parameters that are influenced by the failure events.

#### 2.3.1. Flooding

Flooding is an extreme condition of improper water management where excessive liquid water accumulates within the porous structure of the electrode. A prolonged flooding event impacts the diffusive properties of the electrode and dilutes reactant concentration levels inducing local starvation and sub-stochiometric electrochemical reactions [38]. However, maintaining equilibrated water levels is a challenging task which entails multiple fluid dynamics and is conditioned by several system parameters.

In [39,40], a bifurcation analysis is conducted on a two-phase PEMFC model to determine relevant system parameters that are impacted by flooding. The authors conclude that the accumulation of water in the electrodes compromise the diffusive properties of the gas diffusion layer increasing mass transportation resistances. The authors in [41] construct a fault tree analysis to graphically represent the different causes and symptoms of flooding. An intermediate event of a reduction in electrode porosity is identified as a root cause of water accumulation. Additionally, reference [42] demonstrates that early-stage flooding occurs at the catalyst layer and the gas diffusion layer as water is expelled into the flow channels. Conclusively, it is widely recognized in the literature that the diffusive property of the gas diffusion layer is a reliable indicator of hydration levels in the MEA and is significant in the treatment of flooding.

In the proposed scheme, flooding effects are accurately modelled by incorporating the mass transfer coefficient of the GDL to the fault model. The resulting macroscopic flooding model would consist of four states, three states $x_{1}$$,x_{2}$,$x_{3}$ from the reference macroscopic model augmented with the mass transfer coefficient β following the differential equation
(25)$$\dot{\beta }=0$$

#### 2.3.2. Catalytic Degradation

Catalytic degradation is a continuous irreversible mechanism which disrupts the chemical composition of the catalyst layer inducing loss of ECSA [38]. Among other factors, load cycling, which is characterized by transient current demands, and reactant impurities are major contributors for accelerated catalytic degradation. In [3], the authors demonstrate that rapid transient current demands are not sustained by the intrinsic slow dynamical behavior of mass transportation in PEMFCs leading to local starvation. As a result, suboptimal conditions in the MEA lead to morphological changes in the catalyst layer promoted by mechanisms such as platinum dissolution and platinum oxide formation and reduction.

Failure effects of catalytic degradation on the macroscopic model are defined by the product term $\xi_{CD}=k_{cd}i_{c,o}$, where $i_{c,o}$ is the exchange current density and $k_{CD}$ is the coupling term given in (Equation 17). As a result, Equation (Equation 2) defining stack current density is modified to
(26)$$i_{c}=\xi_{CD}\left(\exp\left(\frac{\alpha_{o}Fx_{3}}{R_{gas}T}\right)-\exp\left(\frac{\left(1-\alpha_{o}\right)Fx_{3}}{R_{gas}T}\right)\frac{x_{2}}{x_{2,0}}\right)$$

The exchange current density $i_{c,o}$ is a catalytic parameter that defines the activeness of the catalyst layer and directly influences the kinematics of electrochemical reactions. Additionally, the system parameter can be utilized as a reliable indicator of a depreciating catalyst layer [43,44].

The resulting macroscopic catalytic degradation model consists of four states, the three states $x_{1}$,$x_{2}$,$x_{3}$ from the original model and the generalized exchange current density $\xi_{CD}$ with the differential equation
(27)$${\dot{\xi }}_{CD}=0$$

In contrast to the multi-scale model, the macroscopic degradation model no longer contains a mechanism for catalytic degradation. While not being able to indicate the precise reason of degradation, the fault diagnosis model is able to indicate any possible degradation mechanism.

### 2.4. Fault Diagnosis Scheme

The proposed fault diagnosis scheme adopts a multiple-model approach operating within a Bayesian framework for model selection [26]. As illustrated in Figure 1, a set of unique macroscopic models (denoted as $x_{M}$) are established each describing a distinctive operating system mode. Each model is associated with a filter that is driven by the system inputs and updated with recorded system measurements. Given the system measurements and prior probabilities of each model, the Bayesian model selection method is implemented to generate fault indication signals carrying information on the current operating regime of the system.

In literature, multiple-model fault diagnosis schemes are widely developed based on a set of models that are distinctively designed to represent a corresponding faulty operating mode. The faulty modes are unique for each model and are incorporated either by deviating relevant system parameters from predetermined nominal values or by injecting sensor and actuator faults [45,46,47]. In this work, a different approach is adopted to achieve model discrimination, as discussed in the following [23].

A set of fault diagnosis models $\left\{M_{0},M_{1},M_{2},...,M_{l}\right\}$ is assumed for the PEMFC system where l is the considered number of faulty operating modes and $M_{0}$ represents a PEMFC system operating under normal conditions. Additionally, the dynamics of the system operating under the *l*th model assumption can be formulated as
(28)$$\begin{array}{cc} {\dot{x}}^{i}\left(t\right) & =f_{i}\left(x^{i}\left(t\right),u\left(t\right),\gamma_{i}\right)+w^{i}\left(t\right)\\ y_{k}^{i} & =h\left(x^{i}\left(t_{k}\right),u\left(t_{k}\right),\gamma_{i}\right)+v_{k}^{i} \end{array}$$
where $x^{i}\left(t\right)$, $u^{i}\left(t\right)$, $y_{k}^{i}$, $w^{i}\left(t\right)$, $v_{k}^{i}$ are, respectively, the state vector, the control input vector, the output vector, the process noise and the measurement noise when the system incorporates the *i*th augmented system parameter. Each model is distinguished by concatenating an internal system parameter $\gamma_{i}$ that is jointly estimated and is attributed to the ith multiplicative fault. As a result, an augmented system state vector is formed as $x^{i}={\left[\begin{array}{cc} x^{T} & \gamma_{i} \end{array}\right]}^{T}$, where $\gamma_{1}=\beta$ in the case of flooding and $\gamma_{2}=\xi_{CD}$ in the case of catalytic degradation, and it is described as a stationary process driven by process noise [27].

#### 2.4.1. Square Root Unscented Kalman Filter

In principle, the main task of Kalman filters is to recursively generate a certain estimate of $x^{i}$ given the history of system measurements $Y_{k}=\left(y_{1}{,y}_{2}\ldots y_{k}\right)$ which is characterized by the posteriori density function $p\left(x^{i}|Y_{k}\right)$. Assuming that the process and measurement noise $w^{i}\left(t\right)∼N\left(0,Q\right)$ and v(t)∼N(0,Q) in (28) are Gaussian and the system dynamics are linear, the Kalman filter parametrizes the posteriori density function $p\left(x^{i}|Y_{k}\right)$ with the mean and the covariance of the distribution and proceeds to solve the state estimation problem. However, if the system is nonlinear, a parametrization method is not applicable and approximations are required. One widely used approach is the Extended Kalman Filter (EKF), which adopts a local linearization method around the predicted mean [48,49]. Among other drawbacks, severe nonlinearities in the system can significantly compromise the performance of the EKF.

To avoid local linearization, the Unscented Kalman Filter and the Square Root Unscented Kalman filter approximate $p\left(x^{i}|Y_{k}\right)$ by an ensemble of deterministically chosen weighted sample points denoted as χ in Algorithm 1 [27,49]. The weights of the sample points $W^{\left(m\right)}$ and $W^{\left(c\right)}$ are defined by the scaling parameters $\lambda =n(\alpha^{2}-1),$$\eta =\sqrt{\left(n+\lambda \right)}$ where *n* is the is dimension of the system state space. In addition, α is used to determine the spread of the sample points around the estimated mean $\hat{x}$ and is set within the range $10^{-4}\leq \alpha \leq 1$. For Gaussian distributions, $\beta_{KF}$ is optimally chosen as $\beta_{KF}\;=2$ [27]. When the methodologically weighted same points are propagated through the nonlinear transformation, denoted by $\varphi \left(.\right)$, the true mean and covariance of the original distribution can be implemented for state estimation [49]. In comparison with UKF, the SRUKF provides better numerical stability by avoiding the computational step of the Cholesky factor at each time iteration. The state estimation algorithm propagates the factor at the initial iteration and implements an orthogonal transformation to compute the covariance matrices [27]. In Algorithm 1, $\varphi \left(.\right)$ denotes the flux of the differential equation system driven by the system input vector u. By implementing Algorithm 1, model-conditioned state estimates ${\hat{x}}_{k}^{i}$ and prior distributions of each model are obtained and propagated to the Bayesian model selection method.
**Algorithm 1** Square root unscented Kalman filter**Require:** Weights for unscented transformation
$$\begin{array}{c} \;\;\;W_{0}^{\left(m\right)}=\lambda /\left(n+\lambda \right)\;\\ \;\;\;W_{0}^{\left(c\right)}=\lambda /\left(n+\lambda \right)+\left(1-\alpha^{2}+\beta_{KF}\right)\;\\ \;\;\;W_{i}^{\left(m\right)}=W_{i}^{\left(c\right)}=1/2\left(n+\lambda \right)\;\;\mathrm{where}\;\;i=1,\ldots ,2n \end{array}$$
 Initialize with
$$\;\;\;\;\;\;\;\;\;{\hat{x}}_{0}=E\left[x_{0}\right],s_{0}=\mathrm{chol}\left\{E\left[\left(x_{0}-{\hat{x}}_{0}\right){\left(x_{0}-{\hat{x}}_{0}\right)}^{T}\right]\right\}$$
 For k∈{1,…,∞} Sigma point calculation and time update
$$\;\;\;\;\;\;\;\;\;\;\chi_{k-1}=\left[\begin{array}{ccc} {\hat{x}}_{k-1} & {\hat{x}}_{k-1}+\eta S_{k} & {\hat{x}}_{k-1}-\eta S_{k} \end{array}\right]$$
$$\;\;\;\;\;\;\;\;\;\;X_{k|k-1}=\varphi \left[\chi_{k-1},u_{k-1}\right]$$
$$\;\;\;\;\;\;\;\;\;\;{\hat{x}}_{k}^{-}=\sum_{i=0}^{2n}W_{i}^{\left(m\right)}X_{i,k|k-1}$$
$$\;\;\;\;\;\;\;\;\;\;S_{k}^{-}=\mathrm{qr}\left\{\left\{\sqrt{W_{1}^{\left(c\right)}}\left(X_{1:2n,k|k-1}-{\hat{x}}_{k}^{-}\right)\;\sqrt{Q}\right]\right\}$$
$$\;\;\;\;\;\;\;\;\;\;S_{k}^{-}=\mathrm{cholupdate}\;\left\{S_{k}^{-},X_{0,k}-{\hat{x}}_{k}^{-},W_{0}^{\left(c\right)}\right\}$$
$$\;\;\;\;\;\;\;\;\;\;Y_{k|k-1}=h\left[X_{k|k-1}\right]$$
$$\;\;\;\;\;\;\;\;\;\;{\hat{y}}_{k}^{-}=\sum_{i=0}^{2n}W_{i}^{\left(m\right)}Y_{i,k|k-1}$$
 Measurement update equations
$$\;\;\;\;\;\;\;\;\;\;S_{\tilde{y}k}=\mathrm{qr}\left\{\left[\begin{array}{cc} \sqrt{W_{1}^{\left(c\right)}}\left[Y_{1:2n,k}-{\hat{y}}_{k}\right] & \sqrt{R} \end{array}\right]\right\}$$
$$\;\;\;\;\;\;\;\;\;\;S_{\tilde{y}k}=\;\mathrm{cholupdate}\;\left\{S_{{\tilde{y}}_{k}},Y_{0,k}-{\hat{y}}_{k},W_{0}^{\left(c\right)}\right\}$$
$$\;\;\;\;\;\;\;\;\;\;P_{x_{kyk}}=\sum_{i=0}^{2n}W_{i}^{\left(c\right)}\left[X_{i,k|k-1}-{\hat{x}}_{k}^{-}\right]{\left[Y_{i,k|k-1}-{\hat{y}}_{k}^{-}\right]}^{T}$$
$$\;\;\;\;\;\;\;\;\;\;K_{k}=\left(P_{x_{k}y_{k}}S_{{\tilde{y}}_{k}}^{-T}\right)S_{{\tilde{y}}_{k}}^{-1}$$
$$\;\;\;\;\;\;\;\;\;\;{\hat{x}}_{k}={\hat{x}}_{k}^{-}+K_{k}\left(y_{k}-{\hat{y}}_{k}^{-}\right)$$
$$\;\;\;\;\;\;\;\;\;\;U=K_{k}S_{\tilde{y}}$$
$$\;\;\;\;\;\;\;\;\;\;S_{k}=\;\mathrm{cholupdate}\;\left\{S_{k}^{-},U,-1\right\}$$


#### 2.4.2. Bayesian Model Selection Method

In this subsection, the Bayesian model selection method is presented to accurately select the underlying model which most likely represents the current operating regime of the system [26]. At first, a prior belief of the likelihood measurement $y_{k}$ conditioned by the *i*-th model is computed as a Gaussian density function
(29)$$\Lambda \left(y_{k}|M_{i}\right)=\frac{1}{{\left(2\pi \right)}^{\frac{n}{2}}\det\left(S_{y_{i}}\right)}\exp\left(-\frac{{\left(y_{k}-{\hat{y}}_{k}^{i}\right)}^{T}S_{y_{i}}\left(y_{k}-{\hat{y}}_{k}^{i}\right)}{2}\right),$$
where ${\hat{y}}_{k}^{i}$ is the predicted measurementand $S_{y_{i}}$ is the measurement estimation error covariance matrix given in Algorithm 1. As proposed in [26,47], interaction probabilities between the multiple models is identified by the elements of a transition matrix with the following characteristics
(30)$$\begin{array}{c} \Pi ={\left\{\pi^{ij}\right\}}_{l\times l^{\prime }}\\ \pi^{ij}=P\left(M\left(k\right)=M_{j}|M\left(k-1\right)=M_{i}\right),\\ \sum_{i=1}^{l}\pi^{ij}=1,0\leq \pi_{ij}\leq 1,\forall i,j\in \left\{1,\ldots ,l\right\}. \end{array}$$

Implementing the transition matrix, a prediction probability of model $M_{i}$ at time instant *k* is computed
(31)$$P\left(M\left(k\right)=M_{i}|y_{k-1}\right)=\sum_{j=1}^{l}\pi^{ji}P\left(M\left(k-1\right)=M_{j}|y_{k-1}\right).$$

Based on the prior belief (Equation 29) and the predicted probability (Equation 31), the Bayes’ Rule is updated to compute the conditional posteriori probability that the model $M_{i}$ is true given the current system measurement $y_{k}$,
(32)$$P\left(M_{i}|y_{k}\right)=\frac{\Lambda \left(y_{k}|M_{i}\right)P\left(M\left(k\right)=M_{i}|y_{k-1}\right)}{\sum_{j=1}^{l}\Lambda \left(y_{k}|M_{j}\right)P\left(M\left(k\right)=M_{j}|y_{k-1}\right)}$$
where the denominator is a normalising constant [23]. On the first iteration, all models are assigned with an equal probability of PMi∣yo=1l. To reliably determine the operating mode of the system a decision logic is derived as,
(33)$$i^{*}=\arg\max_{i}P\left(M_{i}|y_{k}\right)\;\mathrm{and}\;P\left(M_{i^{*}}|y_{k}\right)>\mu_{T}$$

## 3. Results

The performance of the proposed fault diagnosis system is tested using two driving cycles, the LA-92 and the New European driving cycle (NEDC). The algorithm is implemented in the MATLAB 2024a environment. For simulations, the ode15s solver is used. Current profiles of each cycle are shown in Figure 2. The LA-92 driving cycle is a dynamometer driving schedule established by the United States environmental protection agency and is used for Class 3 heavy-duty vehicles with an average speed of 36.74 km/h and a total distance of 17.70 km. The NEDC driving cycle is used for light-duty vehicles with an average speed of 33.6 km/h and a total distance of 11.007 km.

For simplification purposes, it is assumed that a sufficient amount of fuel and air is supplied to sustain the load demands of each driving cycle. Additionally, it is assumed that the vehicle has completed 20 driving cycles and all exhibited simulation results are based on the 21st driving cycle. This assumption is implemented in order to observe the temporal effects of catalytic degradation on cell performance.

Results of each cycle are organized in this section as follows. Firstly, the current profile of each cycle is plotted to demonstrate the transient operating regimes demanded from the stack. Secondly, the robustness and stability of the SRUKF in providing optimal real time state estimates is investigated based on $M_{norm}$. Finally, two failure events of flooding and catalytic degradation are introduced to the PEMFC multi-scale degradation model during the 21st driving cycle. Accordingly, the performance of the proposed fault diagnosis scheme is evaluated in the detection and isolation of each failure event. Including all simulation results of the state estimates provided by the SRUKF for each diagnosis model greatly increases the length of the printed paper. However, the MATLAB 2024a code is available for the reader upon request.

### 3.1. NEDC Driving Cycle

A testing protocol for PEMFCs is established in [50]. It processes the NEDC driving schedule into the newly identified Fuel Cell Dynamic Load Cycle (FCDLC). The conversion is implemented to facilitate the testing of PEMFCs based on a current profile that emulates the NEDC driving cycle and is characterized by the maximum current density of the stack. The top plot in Figure 2 plots the current profile based on a PEMFC with a maximum current of 220 A. The NEDC driving cycle consists of an urban (low-speed) driving schedule followed by a profile emulating highway driving conditions.

In Figure 3, the internal states of a PEMFC system are plotted after sustaining 20 cycles of the FCDLC with a continuously degrading catalyst layer and in response to two failure events. Between 300 s and 500 s, flooding is introduced to the PEMFC system through the parametric change $\beta_{f}$ = 0.6β where 40% of the diffusive properties of the GDL is compromized. To emulate a sudden incident of cell poisoning due to reactant impurities, an induced catalytic degradation is provoked between 800 s and 100 s by varying the forward reaction rate constant in (9) to ${\tilde{k}}_{1}\;=\;10k_{1}$. In addition, state estimates provided by the SRUKF based on the assumptions of $M_{norm}$ are plotted in Figure 3. At the beginning of operation and despite initial condition differences, the SRUKF exhibits stability and robustness properties and achieve convergence to true system states value. Discrepancies between the estimated states and measurements are due to plant–model mismatch between the reference model and $M_{norm}$. Note that $M_{norm}$ assumes a new and fully functional catalyst layer, while the reference model already exhibits symptoms of catalytic degradation and ageing.

In addition, the imposed failure effects of flooding and accelerated catalytic degradation cause further deviations between the two models which the SRUKF cannot compensate. Significantly, the effects of a continuously degrading catalyst layer can be observed by an increase in activation losses $x_{3}$. As experimentally investigated in [51], deformation of the catalyst layer through platinum dissolution and the loss of ECSA increases the local current density at the catalyst level inducing a rise in the activation losses. The failure mechanisms of flooding increase interlayer mass transportation resistances and are indicated by an instantaneous reduction in the molar concentration of oxygen in the catalyst layer.

Figure 4 plots the estimates of the augmented system parameter β in $M_{flood}$ and $\xi_{CD}$ in $M_{CD}$ provided by the SRUKF. Both parameters are initiated with values which correspond to a newly manufactured PEMFC. As observed in the top plot of Figure 4, the estimation of β is achieved with a convergence time of 38 s. The delay is attributed to the low current profile at the initiation of the FCDLC. Low currents produce insignificant differences in the molar concentrations between the gas channels and the catalyst layer, i.e., interlayer flux in (Equation 1) is insufficient, in order for β to be detected by the diagnosis system. However, at 34 s current demands are sufficient to excite mass transportation dynamics in the MEA, and in a range of 4 s, the state estimates converge to the true value of β. During the flooding event induced by $\beta_{f}$, the SRUKF robustly converges and provides correct estimates of the newly defected system parameter value.

The temporal evolution of $\xi_{CD}$ and the corresponding estimates are shown in the bottom plot of Figure 4. After 20 cycles of the FCDLC, the value of $\xi_{CD}$ has depreciated from 0.8 to 0.17, which is the initial value shown at the beginning of the plot. The convergence time to estimate $\xi_{CD}$ is around 35 s, which is an acceptable performance due to the slow temporal failure effects of catalytic degradation. Although the deviation is negligible, the diagnosis system slightly responds to the failure event of flooding during the estimation of $\xi_{CD}$. It is a natural and transient response of the diagnosis system to the detection of an anomaly in the system measurements. Finally, induced catalytic degradation between 800 s and 1000 s is recognized by an increase in the rate of depreciation of $\xi_{CD}$ and is correctly estimated by the diagnosis system.

Fault indication signals carrying knowledge on the current system operating regime generated according to Section 2.4.2 are plotted in Figure 5. As a consequence of the assumption of a continuously degrading fuel cell, $P\left(M_{CD}|y_{k}\right)$ gains the highest probability, indicating the current operating regime of the PEMFC system with catalytic degradation. At 300 s, a flooding event occurs and is instantaneously indicated with a detection delay of less than 20 s. Although the flooding event lasts between 300 s and 500 s, perturbations in the fault indication signals can be observed. A contributing factor is the low-current profiles that generate insufficient interlayer flux that restricts the diagnosis system in accurately estimating β and, consequently, in detecting flooding. Although flooding effects are modelled by an immediate blockage of the gas channels, a gradual slow flooding mechanism may also possible in PEM fuel cells. However, detecting sudden flooding is harder to test for the proposed fault diagnosis system. Additionally, it can be extrapolated that the diagnosis system is also able to detect a gradual flooding mechanism due to the ability of the proposed scheme to detect and diagnose the slow dynamics of catalytic degradation. After purging the stack at 500 s, catalytic degradation is indicated with $P\left(M_{CD}|y_{k}\right)>0.5$. Induced catalytic degradation can be observed as the probability gradient of $P\left(M_{CD}|y_{k}\right)$ slightly increases, detecting an increase in the depreciation rate of the catalyst layer.

### 3.2. LA92 Cycle

The LA92 driving cycle, shown in the bottom plot of Figure 2, is adopted to evaluate the proposed fault diagnosis system for Class 3 heavy-duty vehicles. However, the following modifications are implemented to the original cycle while maintaining the main features and characteristics of the driving schedule. At first and in accordance with [50], open-circuit voltage conditions are substituted with 5% of the maximum power demanded by the cycle. Additionally, assuming that the truck is equipped with several stacks, as in the case of the Hyundai Xcient truck, the maximum power required by the driving cycle for a single stack is limited to 100 kW. Implementing the fuel cell vehicle model introduced in Section 2 with the previously mentioned assumptions, the current profile of the modified LA92 driving cycle is shown in the bottom plot of Figure 2. It can be observed that the LA92 driving cycle imposes significantly more aggressive conditions than the NEDC driving cycle.

In Figure 6, the internal states of a continuously degrading PEMFC system are plotted after delivering power to 20 cycles of the LA92 and in response to two failure events. Both failure events are identical in specifications to the NEDC driving cycle. However, flooding occurs between 500 s and 750 s and an accelerated degradation mechanism occurs between 1000 and 1250 s. At the beginning of the cycle, the SRUKF exhibits high stability, and convergence is achieved despite initial conditions differences. Failure effects of flooding and catalytic degradation can be analyzed through the plant-model mismatch between the measurements and the state estimates of $M_{norm}$. Specifically, the assumption of a continuously degrading catalyst layer through platinum dissolution is indicated by the increase in activation losses $x_{3}$. Additionally, flooding is observed by an immediate drop in the mass concentration levels of oxygen in the catalyst layer $x_{2}$.

The true system state values and the estimates of the system parameters β defined in $M_{flood}$ and $\xi_{CD}$ in $M_{CD}$ are plotted in Figure 7. A convergence delay of 67 s is observed in the estimation of β due to low current profiles at the beginning of the cycle. Subsequently, insufficient mass transportation dynamics hinders the diagnosis system in correct estimating β. However, during the flooding event between 300 s and 500 s, correct estimation is achieved of the faulty system parameter. After 20 cycles of the LA92, the value of $\xi_{CD}$ depreciates from 0.8 to 0.26.

It is noteworthy that the degradation rate in the NEDC cycle is higher than the LA92. An explanation for the different catalytic depreciation values can be attributed to the current profile of the FDCLC which demands extended operation times at lower currents. As a result, low current densities indicate higher potential differences which provide optimal conditions to initiate the electrochemical reactions of platinum dissolution and platinum oxide formation (Equation 7)–(Equation 8) During the flooding event, a slight deviation in the estimates of $\xi_{CD}$ can be observed. Accurate estimation of $\xi_{CD}$ is achieved during the accelerated catalytic degradation in the duration between 800 s and 1000 s.

In Figure 8, the Bayesian model selection method (Equation 29)–(Equation 33) is implemented to diagnose flooding and catalytic degradation. Although flooding occurred between 500 s and 750 s, fault detection was only achieved at 575 s. The 75 s detection delay time is attributed to the low currents that are insufficient to initiate significant interlayer flux between the flow channels and the catalyst layer. At 570 s, the vehicle accelerates drawing higher current densities from the stack and within 5 s $P\left(M_{flood}|y_{k}\right)>0.5$, and flooding is detected by the diagnosis system. After the cell is purged, $P\left(M_{CD}|y_{k}\right)$ gains the highest probability indicating catalytic degradation.

Despite the plant–model mismatch imposed by excluding the microscopic sub-model from the fault diagnosis system, the multiple-model approach combined with SRUKF effectively detects catalytic degradation. This signifies that the proposed scheme is not only limited to detecting the degradation mechanisms of platinum dissolution and platinum oxide formation, but may identify other mechanisms that induce catalytic degradation. Another advantage offered by the proposed approach is the ability to perform online fault diagnosis of numerous faults by transitioning between the set of models and selecting the correct model indicating the current operating regime of the system. However, the scheme only classifies predetermined faults. If a non-predetermined fault occurs, the scheme may detect an error by assigning low probabilities to all fault models, but a classification of the error will no longer be possible.

## 4. Conclusions

Flooding and catalytic degradation are classified as critical cell failures that directly impact PEMFC durability. In this paper, a multiple-model fault diagnosis approach is adopted for the early detection of flooding and catalytic degradation. In addition to a model representing normal PEMFC operating regime, the multiple-model scheme consists of two faulty operating models. A model representing flooding conditions is established by incorporating the mass transfer coefficient of the GDL as an additional system state. Similarly, a model capturing describing catalytic degradation is developed by augmenting the exchange current density into the state vector. Both parameters are jointly estimated using the square root unscented Kalman filter.

A diagnostic oriented multi-scale degradation model is developed as a testing environment for the proposed fault diagnosis scheme. The multi-scale model couples the molecular catalytic degradation mechanisms of platinum dissolution and platinum oxide formation with global cell performance. Finally, the diagnosis scheme is evaluated using the FCLDC which is an extension of the NEDC driving cycle and the LA 92 driving cycle. Results demonstrate that both faulty events are successfully detected and isolated by the fault diagnosis system. However, the detection of flooding is only possible at higher current densities and can be undetected if the current demands are too low. Additionally, different catalytic degradation rates are identifiable by the fault diagnosis system.

The presented fault diagnosis scheme can be integrated into vehicular online supervision systems to effectively manage and plan mitigation strategies to avoid flooding and catalytic degradation. Estimating the internal states of the PEMFC systems, for instance, the catalytic qualities of the stack, can also be used. The sustems can act as indicators for preventive maintenance actions to avoid irreversible damages to other components of the stack.

## Figures and Tables

**Figure 1 sensors-25-00029-f001:**
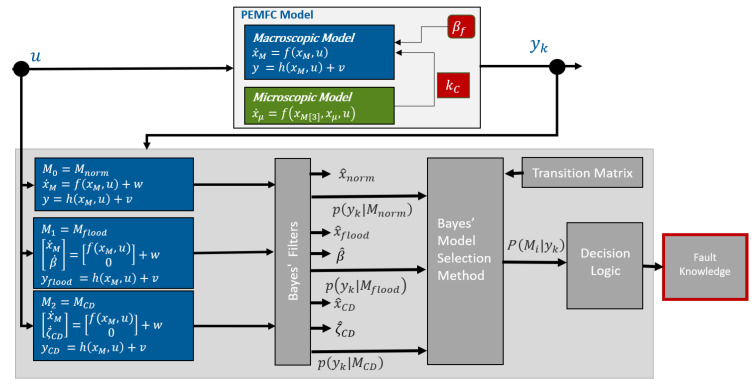
At the top of the figure is the multi-scale model, which is considered as the plant, driven by the system inputs *u* and observed measurement $y_{k}$. The multiple-model fault diagnosis scheme is highlighted in the gray box. In the proposed algorithm, the Bayes’ filter implemented is the square root unscented Kalman filter.

**Figure 2 sensors-25-00029-f002:**
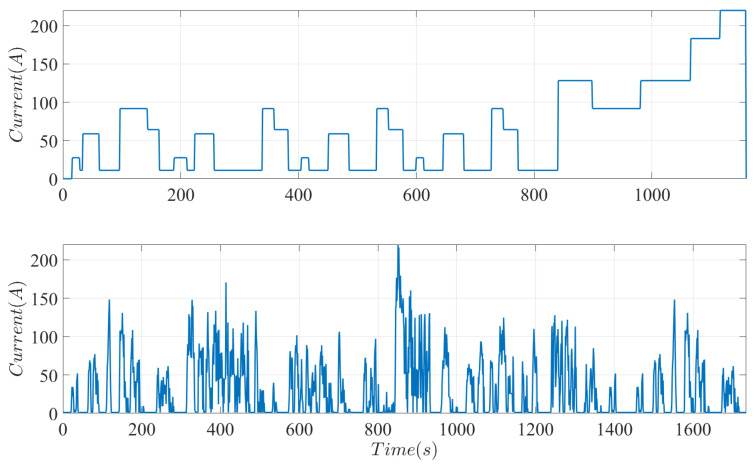
Fuel cell dynamic load cycle established in [50] to emulate driving conditions of the NEDC driving cycle (**top** plot). Modified LA92 driving cycle with no open circuit voltage conditions and a maximum power required of 100 kW (**bottom** plot).

**Figure 3 sensors-25-00029-f003:**
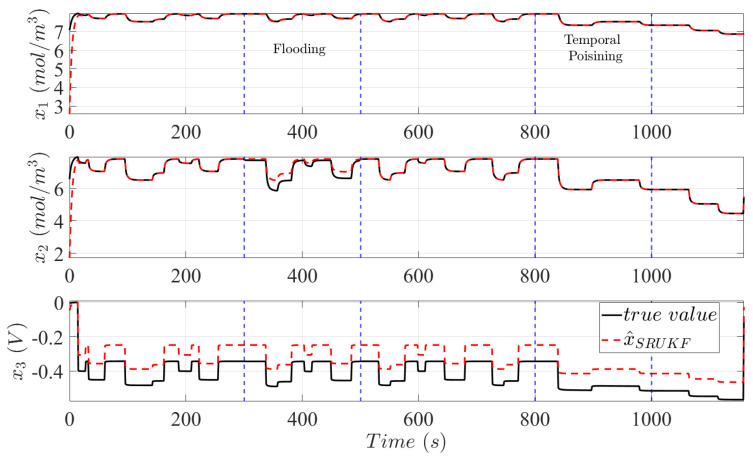
The black lines are the true system states value during an NEDC cycle and the red dashed line are the state estimates provided by the SRUKF based on $M_{norm}$. Observed discrepancies are due to deliberate plant-model mismatch caused by the induced failure mechanisms of flooding and catalytic degradation. Vertical blue dotted lines indicate the duration of imposed system failures.

**Figure 4 sensors-25-00029-f004:**
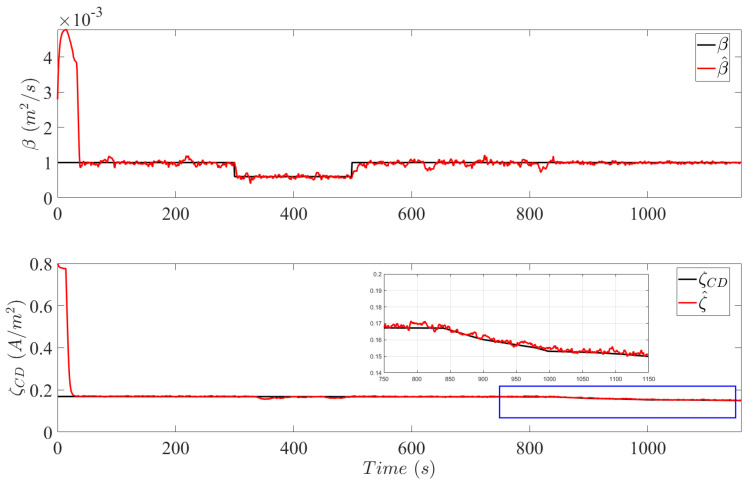
Estimation of system parameter β in $M_{flood}$ (**top** plot) and $\xi_{CD}$ in $M_{CD}$ (bottom plot). A magnified section, illustrated by a blue box in the **bottom** plot, is shown to observe the effects of accelerated catalytic degradation due to temporal poisoning.

**Figure 5 sensors-25-00029-f005:**
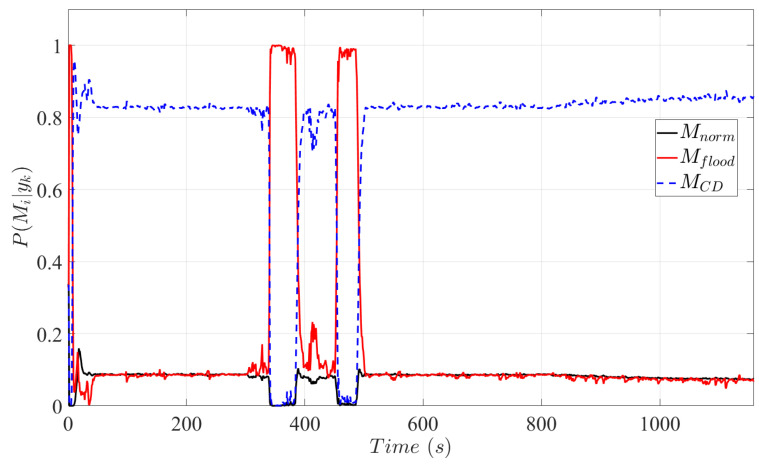
Probabilities based on the Bayesian model selection method indicating the operating regime of the system.

**Figure 6 sensors-25-00029-f006:**
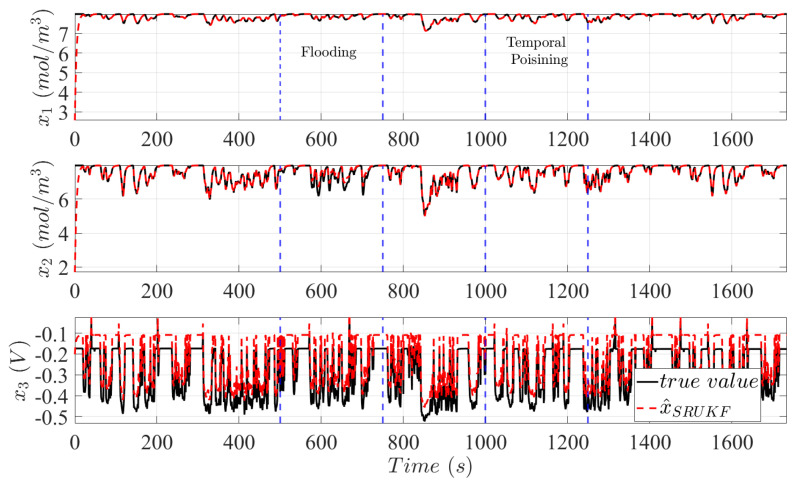
Black line plots the true system states according to the LA92 driving cycle and the red dashed line are the state estimates provided by the SRUKF based on $M_{norm}$. Discrepancies between the estimates and measurements are due to plant-model mismatch. Blue dotted lines indicate the duration of a flooding event between 500 s and 700 s and an accelerated catalytic degradation event between 1000 s and 1300 s.

**Figure 7 sensors-25-00029-f007:**
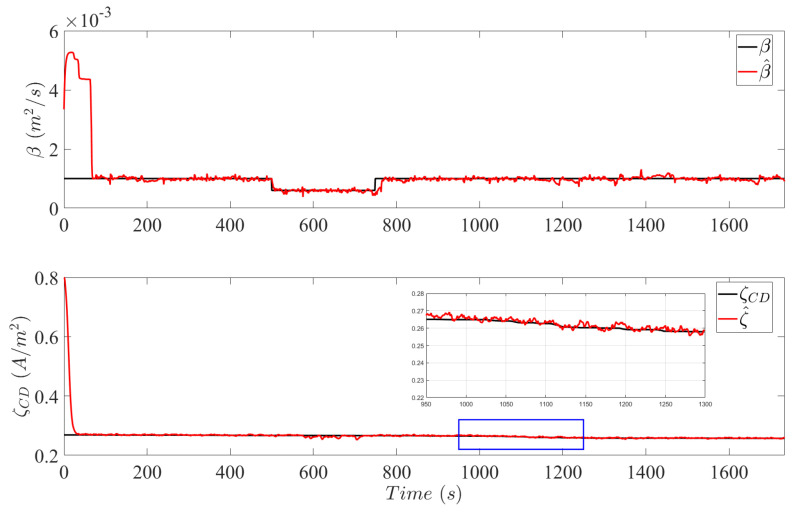
Estimation of system parameter β in $M_{flood}$ (**top** plot) and $\xi_{CD}$ in $M_{CD}$ (**bottom** plot). The blue box incorporates a magnified section of the **bottom** plot to observe the effects of accelerated degradation on the predefined catalytic parameter defined in Section 2.3.2.

**Figure 8 sensors-25-00029-f008:**
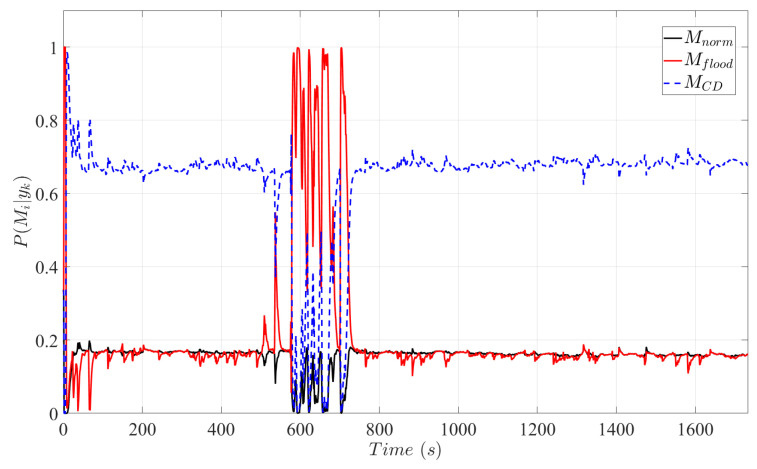
Probabilities based on the Bayesian model selection method.

## Data Availability

Data are contained within this article.

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
