# Peer review of "Square Root Unscented Kalman Filter-Based Multiple-Model Fault Diagnosis of PEM Fuel Cells"

_sensors, 2024, doi:10.3390/s25010029_

Round 1

Reviewer 1 Report

Comments and Suggestions for Authors

This manuscript developed a diagnostic oriented multi-scale PEMFC catalytic degradation model, and a model-based Kalman filter observer is employed for the early detection of water management and degradation. However, the following questions should be solved before possible publication.

1.    Purpose and significance of the study is unclear. Please incorporate a discussion in the introduction or add “Problem Formulation” section to elaborate it.

2.    How many dimensions is the diagnostic oriented model? Seemed that only state variables in some key points are considered, it will be better to demonstrate the model reliability.

3.    In “Microscopic Model” subsection, please elaborate on the reasonability of “two states” and “worst-case degradation scenario” simplifications.

4.    The two-phase flow is generally considered with large time constant, resulting in long-time dynamic procedure of flooding, thus the dynamic analysis for flooding is kindly suggested.

5.    The presentation of mathematical formulas needs to be more standardized. On page 6, line 263, ‘H+’ should be ‘H+’. On page 9, line 357, ‘e’ is better expressed as ‘exp’. Please recheck all the equations in the manuscript.

Author Response

We kindly thank the reviewer's comments.  Please find the response in the attached file

Reviewer 2 Report

Comments and Suggestions for Authors

1. The experimental results are not detailed enough and don't clearly show the running process of the algorithm.

2. The calculation process of the model is not demonstrated in detail.

3.The experiment lacks the comparison  with existing recent algorithms.

Comments on the Quality of English Language

no

Author Response

(The authors gave the same response as above.)

Reviewer 3 Report

Comments and Suggestions for Authors

Authors developed a fault diagnosis model for early detection of flooding and catalytic degradation in PEMFC. The proposed method is evaluated using realistic driving cycles, demonstrating its potential to improve PEMFC management and supervision in electric propulsion systems. I have some minor issues, which may help to further improve the manuscript.

1.     The proposed method could be successfully used to detect flooding and catalytic degradation, while there could be other unexpected conditions during operation, would the proposed method still be effective?

2.     It would be helpful for authors to include more detailed introduction to the proposed PEMFC algorithm and highlight its strength in state estimation and model selection.

3.     Authors highlighted the diagnostic-oriented multi-scale PEMFC catalytic degradation model. It would be beneficial to further explain how the multi-scale model more comprehensively captures the relationship between PEMFC internal dynamics and global stack performance, and how this integrated perspective aids in early fault detection.

4.     It could be better to further emphasize the implications of the results for practical applications and discuss potential limitations of the research and future directions for academic study and industrial application.

Author Response

(The authors gave the same response as above.)

Round 2

Reviewer 1 Report

Comments and Suggestions for Authors

This manuscript has been appropriately revised, and I think it meets the requirements for publication.

Reviewer 2 Report

Comments and Suggestions for Authors

the paper can be accepted